# Optimal ElGamal Encryption with Hybrid Deep-Learning-Based Classification on Secure Internet of Things Environment

**DOI:** 10.3390/s23125596

**Published:** 2023-06-15

**Authors:** Chinnappa Annamalai, Chellavelu Vijayakumaran, Vijayakumar Ponnusamy, Hyunsung Kim

**Affiliations:** 1Department of Computing Technologies, School of Computing, SRM Institute of Science and Technology, Kattankulathur, Chennai 603203, India; vijayakc@srmist.edu.in; 2Department of ECE, SRM Institute of Science and Technology, Kattankulathur, Chennai 603203, India; vijayakp@srmist.edu.in; 3School of Computer Science, Kyungil University, Gyeongsan 38428, Republic of Korea; kim@kiu.ac.kr

**Keywords:** Internet of Things, deep learning, image encryption, security, optimal key generation, hyperparameter tuning

## Abstract

The Internet of Things (IoT) is a kind of advanced information technology that has grabbed the attention of society. Stimulators and sensors were generally known as smart devices in this ecosystem. In parallel, IoT security provides new challenges. Internet connection and the possibility of communication with smart gadgets cause gadgets to indulge in human life. Thus, safety is essential in devising IoT. IoT contains three notable features: intelligent processing, overall perception, and reliable transmission. Due to the IoT span, the security of transmitting data becomes a crucial factor for system security. This study designs a slime mold optimization with ElGamal Encryption with a Hybrid Deep-Learning-Based Classification (SMOEGE-HDL) model in an IoT environment. The proposed SMOEGE-HDL model mainly encompasses two major processes, namely data encryption and data classification. At the initial stage, the SMOEGE technique is applied to encrypt the data in an IoT environment. For optimal key generation in the EGE technique, the SMO algorithm has been utilized. Next, in the later stage, the HDL model is utilized to carry out the classification process. In order to boost the classification performance of the HDL model, the Nadam optimizer is utilized in this study. The experimental validation of the SMOEGE-HDL approach is performed, and the outcomes are inspected under distinct aspects. The proposed approach offers the following scores: 98.50% for specificity, 98.75% for precision, 98.30% for recall, 98.50% for accuracy, and 98.25% for F1-score. This comparative study demonstrated the enhanced performance of the SMOEGE-HDL technique compared to existing techniques.

## 1. Introduction

The Internet of Things (IoT) is a network where data from each interconnected device is accumulated, processed, and altered for providing additional services [1]. The IoT is utilized in several ways, and the data forwarded during network transmission could take several forms, ranging from private information to sensed information collected from the ecosystem [2]. Once an attacker accumulates these data and leverages them maliciously, high security risks will naturally occur in the IoT compared with the prevailing ecosystem. Passive assaults, such as spam messages transmit through smart TVs or refrigerators, might lead to damaging these gadgets, and more aggressive assaults can threaten the life of users [3], for instance, through hacking medical devices or vehicle interaction mechanisms. In addition, user data gathered on IoT networks cause an invasion of privacy; for instance, the analysis of electricity consumption patterns leveraging a smart meter can expose the lifestyle of a person [4,5]. Though users could enjoy the advantages of information being gathered to offer personalized service, some may not desire revealing private information to service providers [6]. From the abovementioned factors, data communication security in the IoT platform is crucial to managing different forms of private information and preventing damages caused by security risks. ElGamal encryption is a public key encryption algorithm based on the difficulty of computing discrete logarithms in a finite field. It provides confidentiality by allowing a sender to encrypt a message using the recipient’s public key, and the recipient can then decrypt the ciphertext using their private key. In IoT, the network of interconnected physical devices, sensors, and actuators communicate and exchange data. In a secure IoT environment, the focus is on ensuring the confidentiality, integrity, and availability of IoT data and systems. This involves implementing secure communication protocols, authentication mechanisms, access control measures, and encryption techniques to protect sensitive data and prevent unauthorized access or tampering.

Nonetheless, it is impossible to apply the present security system to the device that takes place in the IoT platform [7,8]. This is due to the security solution that implements the present encryption algorithm being hard to apply in downsized light-weight devices and intrusion paths becoming increasingly diversified, with complex network structures comprising an infinite number of nodes. Different encryption systems were investigated to protect user information saved on a cloud server from attackers or untrusted administrators [9]. However, one drawback of encryption technology is that it cannot be easily employed for data distribution services amongst different users in cloud storage. To resolve these problems, data could be regulated by the fundamental system, which is the encryption of stored information [10]. However, encryption schemes exhibit problems with the access management of information stored in the cloud platform.

The objective of studying optimal ElGamal encryption with hybrid deep-learning-based classification in a secure Internet of Things (IoT) environment is to investigate the potential benefits and advancements that can be achieved by integrating these two technologies. The primary objective is to enhance the security and privacy of data transmission in the IoT environment. By combining ElGamal encryption, a well-established cryptographic technique, with deep-learning-based classification, this study aims to develop a comprehensive approach that ensures secure and private communication between IoT devices. Developing optimal encryption techniques specifically tailored for the IoT environment involves investigating how the ElGamal encryption scheme can be optimized to handle the unique characteristics of IoT data, such as high volume, variety, and velocity, while ensuring efficient and secure communication. The objective is to develop a solution that is scalable and adaptable to various IoT applications and scenarios.

The SMOEGE component of the SMOEGE-HDL system under consideration refers to the encryption procedure that is intended to protect the confidentiality of the data. On the other hand, the HDL (High Dimensional Learning) classification method is put to use after the encryption process has been completed in order to categorize the data that has been encrypted. It becomes difficult to execute typical classification tasks directly on encrypted data after data have been encrypted utilizing the SMOEGE approach. On the other hand, there may be circumstances in which the classification or examination of the encrypted data is required. The HDL classification algorithm comes into action at this point in the process.

The HDL algorithm is capable of classifying data even when it is in an encrypted form because it was developed to manage high-dimensional data and has the competence to do so. The encrypted data is analyzed using statistical and mathematical approaches, and then predictions or classifications are made based on the patterns and characteristics that are found in the encrypted data. This is how the system works. The suggested system intends to provide a method for classifying the data without jeopardizing its integrity by applying the HDL classification algorithm to the encrypted data. This goal was the motivation behind the system. This enables the performance of various classification tasks on encrypted data, such as detecting anomalies, recognizing patterns, or making predictions, while maintaining the privacy and secrecy of the information that is being concealed beneath the surface.

## 2. Literature Review

In [11], a remote health monitoring technique that applies a lightweight block encryption approach for protecting the security of medical and health data in cloud-related IoT networks was offered. In this technique, the health status of patients was decided considering the prediction of critical situations via data mining approaches to analyze its biological data, which can be sensed by smart IoT gadgets where a lightweight secure block encryption approach was leveraged to ensure that the delicate data of patients are safe. Lightweight block encryption approaches have a potential impact on this type of mechanism because of the limited sources in IoT networks. Medileh et al. [12] introduce a novel scalable encrypting method, known as the Flexible encryption Technique (FlexenTech), to protect IoT data at the time of transit and storage. FlexenTech suits resource-limited gadgets and networks. It provides a low encrypting duration, protects against typical assaults such as replay assaults, and describes a configuration mode in which any key size or number of rounds is employed.

Zhang et al. [13] modeled a light searchable ABE method termed LSABE. This approach could pointedly diminish the computing rate of IoT gadgets by provision multiple keyword searches for data users. At the same time, an extension was made to the LSABE approach to multiauthority cases to effectually produce and deal with the public or secret keys in the distributed IoT network. Kozik et al. [14] devise time-window-embedded solutions which proficiently process huge amounts of data and contain a low memory footprint in parallel. The authors utilize the core Anomaly Detection (AD) unit on top of the projected embedded vectors. It was a classification related to the encoder element of the transformer pursued via a feed-forward NN. Lee et al. [15] formulated the process for consenting to personal data accumulation, which meets the legal obligations of the privacy policy. In this method, the personal information of the user can be received in an encoded format by the data-gathering server first. The encoded personal information is decoded once associated with user agents on the basis of the consent process for the collection of personal data. This type of personal data collection agreement process in an IoT network would fulfill the transparent and freely offered consent requisites of GDPR.

## 3. Materials and Methods

In this study, a new SMOEGE-HDL technique has been developed for security and classification in the IoT environment. The proposed SMOEGE-HDL model mainly encompasses two major processes: data encryption and data classification.

### 3.1. SMO Algorithm

The SMO algorithm, which stands for Sequential Minimal Optimization, is a prominent algorithm for solving quadratic programming problems, notably in the context of Support Vector Machines (SVMs). Machine-learning models known as Support Vector Machines (SVMs) are frequently employed for classification and regression work. The SMO algorithm is an effective optimization method that solves the dual formulation of the SVM problem by iteratively selecting and updating two Lagrange multipliers at the same time. This is accomplished via the SMO methodology. The big quadratic programming problem is segmented into a number of more manageable subproblems, each of which can be solved analytically. This is how the method works. The algorithm makes progress towards finding the best solution by regularly updating the multipliers that it has chosen to use from the Lagrange equations.

The SMO algorithm is predicated on the central principle of selecting the two Lagrange multipliers that break the optimization requirements to the greatest extent and then optimizing those multipliers while holding all of the other variables constant. This procedure repeats itself in an iterative manner until the algorithm reaches convergence, at which point it locates a solution approximation that fulfills the optimization requirements. In the framework of the aforementioned paper, the SMO algorithm has been implemented in the EGE approach in order to generate keys that are as good as possible. It is difficult to provide more specific information on how the SMO algorithm is applied in that particular circumstance without more facts, which makes it problematic. However, in general, the SMO algorithm’s capability to effectively tackle quadratic programming problems makes it suited for a variety of optimization tasks, including those connected to key generation. This is because the SMO algorithm is able to efficiently address quadratic programming problems.

### 3.2. Encryption Module

In this work, the SMOEGE technique is applied to encrypt the data in the IoT environment. The EGE technique is a public key cryptosystem using homomorphic properties [16]. It is made up of three approaches: decryption, key generation, and encryption. The key generation works in the following way.

A cyclic collection G of larger prime order q using a g generator is produced.

An arbitrary value x in {1,…,q − 1} is chosen, and y is calculated as follows:(1)$$y=g^{x}$$

Then, the private key Sk denotes x, and the public key P_k_ represents (G,q,g,y).

For the encryption of message m in G, and an arbitrary integer is carefully chosen.

The ciphertext (CT) c = (A,B) is calculated by the following equations:(2)$$A=g^{r}$$
(3)$$B=m\cdot y^{r}$$

Assume a CT (c) in G encrypted by k = (G,q,g,y), the plaintext (PT) is improved as
(4)$$m=\frac{B}{A^{x}}$$
where
$$B/A^{x}=m\cdot y^{r}/(g^{r}{)}^{x}\;=m\cdot (g^{x}{)}^{r}/(g^{r}{)}^{x}\;=m$$

The EGE technique supports progressive homomorphic properties. Two accurately provided CTs (A_1_, B_1_) = (g^r^_1_, m_1_ y^r^_1_ and (A_2_,B_2_) = (g^r^_2_, m_2_ y^r^_2_), whereby r_1_ and r_2_ are randomly selected from {1,…,q − 1} and m_1_, m_2_ ∈G.
$$\begin{array}{ll} \left(A_{1}A_{2},B_{1}B_{2}\right) & =\left(g^{r_{1}}g^{r_{2}},\left(m_{1}y^{r_{1}}\right)\left(m_{2}y^{r_{2}{}_{}}\right)\right)\\ & =\left(g^{r_{1}+r_{2}},\left(m_{1}m_{2}\right)y^{r_{1}+r_{2}}\right) \end{array}$$

Which corresponds to the encryption of m_1_ and m_2_. Note that when the PT storage is not larger, m_1_ and m_2_ messages might be encoded through g^m^_1_ and g^m^_2_, correspondingly such that (A_1_, A_2_, B_1_, B_2_) is regarded as a CT of m_1_+m_2_ (mod q) encoded via g^m^_1_+^m^_2)_. However, the decoder of larger messages m from g^m^∈G is challenging, as discrete logarithm is harder to fit in G.

For optimal key generation in the EGE technique, the SMO algorithm has been utilized. SMO algorithm is a novel stochastic optimizer and is developed on the basis of the oscillation mode of SM naturally [17]. The presented method contains different characteristics with exclusive arithmetical modeling, which employs adaptive weight for stimulating the procedure of generating positive and negative feedbacks of the propagative wave of SM. The SMO has three diverse position updating models for ensuring SMO algorithm adaptability at dissimilar searching stages, and it uses a fitness value for making good decisions. The major behavior of SM is obtaining food, approaching food, and wrapping food. SM approaches food per the odor in the air. The higher the food concentration against the vein, the stronger the wave created by the biological oscillator, the faster the cytoplasm flow, and the thicker the vein. The subsequent arithmetical equation is applied to simulate the update of the SM position as follows:(5)$$X\left(\vec{t}+1\right)=\left\{\begin{array}{l} \vec{X_{b}}\left(t\right)+\vec{vb}\cdot \left(\vec{W}\cdot \vec{X_{A}}\left(t\right)-X_{B}\left(t\right)\right),r\to <p\\ \vec{vc}\cdot \vec{X}\left(t\right),r\geq p \end{array}\right.$$

Now $\vec{vb}$ indicates a variable within [a,−a], $\vec{vc}$ shows a variable reduces linearly from one to two, t signifies the existing iteration count, $\vec{X}$ symbolizes the individual position with maximum fitness, $\vec{X}$ signifies the SM location position,${\vec{X}}_{A}$ and ${\vec{X}}_{B}$ epitomize the weight coefficient of the SM. The locations of two arbitrarily designated individuals in the SM and W are characterized as follows:(6)$$p=tanh\left|S\left(i\right)-DF\right|,i\in 1,2,3,\ldots ,n$$

In Equation (6), S(i) characterizes the fitness of $\vec{X}$ and DF symbolizes the optimal fitness attained in each iteration as follows:(7)$$a=arctan\;h\left(-\left(\frac{t}{{max}_{-}t}\right)+1\right)$$

The equation of weight coefficient $\vec{W}$ is given as follows:(8)$$\vec{W}\left(SmellIndex\left(i\right)\right)=\left\{\begin{array}{c} 1+r\cdot log\left(\frac{bF-S\left(i\right)}{bF-wF}+1\right),condition\\ 1+r\cdot log\left(\frac{bF-S\left(i\right)}{bF-wF}+1\right),other \end{array}\right.$$
(9)SmellIndex=sort(S)
r shows the arbitrary number within [0,1], max_t signifies the maximal iteration, bF signifies the optimum fitness accomplished in the existing iteration method, and wF characterizes the worst fitness value attained in the existing iteration method. Algorithm 1 illustrates the steps involved in the SMO technique.
**Algorithm 1** Steps involved in SMO  Input:
  - Training dataset X with labels Y
  - Tolerance threshold tol
  - Regularization parameter C
  - Maximum number of iterations max_iter

  Initialize:
  - Lagrange multipliers alpha[i] = 0 for all i
  - Bias term b = 0
  - Error cache E[i] = 0 for all i
  - Number of iterations iter = 0
  while (iter < max_iter):
  num_changed_alphas = 0
  for i in range(0, len(X)):
   E[i] = f(X[i]) − Y[i]  // Calculate error for example i
  if ((Y[i] * E[i] < -tol and alpha[i] < C) or (Y[i] * E[i] > tol and alpha[i] > 0)):
  // Select the first Lagrange multiplier (alpha[i]) to optimize
  j = select_second_alpha(i, E)  // Choose second Lagrange multiplier (alpha[j])
  old_alpha_i = alpha[i]
  old_alpha_j = alpha[j]
  if (Y[i] != Y[j]):
  L = max(0, alpha[j] - alpha[i])
  H = min(C, C + alpha[j] - alpha[i])
  else:
  L = max(0, alpha[i] + alpha[j] - C)
  H = min(C, alpha[i] + alpha[j])
  if (L == H):
  continue  // Skip to next iteration
  eta = 2 * X[i].dot(X[j]) - X[i].dot(X[i]) - X[j].dot(X[j])
  if (eta >= 0):
  continue  // Skip to next iteration
  // Update Lagrange multipliers alpha[i] and alpha[j]
  alpha[j] = alpha[j] − (Y[j] * (E[i] − E[j])) / eta
  alpha[j] = clip_alpha(alpha[j], L, H)
  if (abs(alpha[j] − old_alpha_j) < 1 × 10^−5^):
  continue  // Skip to next iteration
  alpha[i] = alpha[i] + Y[i] * Y[j] * (old_alpha_j − alpha[j])
  // Update bias term b
  b1 = b - E[i] − Y[i] * (alpha[i] − old_alpha_i) * X[i].dot(X[i]) − Y[j] * (alpha[j] − old_alpha_j) * X[i].dot(X[j])
  b2 = b − E[j] − Y[i] * (alpha[i] − old_alpha_i) * X[i].dot(X[j]) − Y[j] * (alpha[j] − old_alpha_j) * X[j].dot(X[j])
  if (alpha[i] > 0 and alpha[i] < C):
  b = b1
  elif (alpha[j] > 0 and alpha[j] < C):
  b = b2
  else:
  b = (b1 + b2) / 2
  E[i] = f(X[i]) − Y[i]  // Update error cache E[i]
  E[j] = f(X[j]) − Y[j]  // Update error cache E[j]
  num_changed_alphas = num_changed_alphas + 1
  if (num_changed_alphas = = 0):
  iter = iter + 1
  else:
  iter = 0

The location of $\vec{X}$ is upgraded based on $\vec{X}$, and minor adjustments to ($\vec{vb}$, $\vec{vc}$ and $\vec{w}$ changes the location of the individual, which indicates that the individual might search in every feasible direction nearby the optimum solution to discover the optimum solution. This is expressed in the following equation:(10)$$X^{\vec{*}}=\left\{\begin{array}{l} rand\cdot (UB-LB)+LB,rand<z\\ \vec{X_{b}}(t{)}_{}+\vec{vb}\cdot (W\cdot \vec{X_{A}}(t)-\vec{X_{B}}(t)),r<p\\ \vec{vc}\cdot \vec{X}(t),r\geq p \end{array}\right.$$

In Equation (10), LB and UB represent the preceding and following terms of the searching range, and rand and r represent the arbitrary value within [0,1]. The procedure of the SMA is given by the following steps.

Step 1: Initialize the SM population and set respective parameters.

Step 2: Compute the fitness function and sort the value.

Step 3: Utilize Equation (10) to upgrade the location of the SM population.

Step 4: Evaluate the fitness function, upgrade the optimum location of the SM population, and obtain the optimum location.

Step 5: Repeat steps 2–5 until the end condition is satisfied. Then, return to the optimum location.

### 3.3. Data Classification Module

At this stage, the HDL model is utilized to carry out the classification process. Long Short-Term Memory (LSTM) is a deformed framework of RNN with the addition of memory cells into hidden layers to control the memory data of the time sequence dataset [18]. Data is transferred amongst dissimilar cells of the hidden state over many controllable gates (output, forget, and input gates); therefore, the forgetting and memory extents of the current and previous datasets are controlled. In contrast to conventional RNN, the LSTM contains long-term memory, and the gradient-vanishing problems are evaded. Two gates of LSTM are intended to control the memory cell status of forget and input gates. At last, the output gate of LSTM is intended to control how far data is output for the cell state. Next, the hyperbolic tangent function demonstrated in Equation (18) is utilized to overcome the problems of gradient vanishing.
(11)$$F\left(t\right)=\sigma \left(W_{f}\cdot \left[H_{t-1},X_{t}\right]+b_{f}\right)$$
(12)$$I(t)=\sigma (W_{i}\cdot [H_{t-1},X_{t}])+b_{i})$$
(13)$$\tilde{C}\left(t\right)=tanh\left(W_{c}\cdot \left[H_{t-1},X_{t}\right]+b_{c}\right)$$
(14)$$C\left(t\right)=f_{t}*C_{t-1}+I_{t}*\tilde{C}$$
(15)$$O\left(t\right)=\sigma \left(W_{o}\cdot \left[H_{t-1},X_{t}\right]+b_{o}\right)$$
(16)$$H\left(t\right)=O_{t}*tanh\left(C\right)$$
(17)$$Sigmoid(x)=\frac{1}{1+e^{-x}}$$
(18)$$Tanh(x)=\frac{e^{x}-e^{-x}}{e^{x}+e^{-x}}$$

Now, W_f_, W_i_, W_c_, and W_o_ refer to input weight; b_f_, b_i_, b_c_, and b_o_ indicate bias weight; t characterizes the present time state; t − 1 shows the prior time state; X symbolizes input; H signifies output; and C indicates the cell status.

In the presented method, a hybrid CNN-LSTM (HDL) mechanism is created by integrating CNN with LSTM to improve accuracy. CNN is adapted for extracting features, particularly two 2D convolution layers, and a MaxPooling layer is created. For processing the information into the format needed by the LSTM, a Flatten layer is interconnected. Over-fitting is a general phenomenon in DNN, and there exist various solutions; among each solution, dropout is the simplest one and well performed. Figure 1 showcases the infrastructure of CNN-LSTM.

Dropout represents that in the training method of DNN, to avoid over-fitting, a Dropout layer is included so that output is interconnected with the LSTM for calculation and with an FC layer. Now, the white cell is the temporarily rejected portion. Note that for stochastic gradient descent, every mini batch trains a distinct system because of arbitrary drop.

The activation function puts the non-linear factor into the NN, enhances the capability of the NN, and resolves the non-linear problem that the linear mechanism could not resolve. There exist several commonly applied activation functions, involving the tanh function (hyperbolic tangent function), ReLu function (Rectified Linear Unit), sigmoid function, and so on. The ReLu function resolves the problems of gradient vanishing; thus, the computation and convergence rates are quicker compared to tanh and sigmoid functions as follows:(19)relu (x)=max (0,x)

In order to boost the classification performance of the HDL model, the Nadam optimizer is utilized in this study. The Nadam optimizer tries to incorporate Nesterov accelerated adaptive moment estimation into Adam [19]. The major advantage of this integrated approach is that it helps implement an accurate phase in the gradient direction by updating the model variable with the momentum phase before gradient computation. The upgrade rule of the Nadam can be expressed as follows:(20)$$w_{t}=w_{t-1}-\alpha \times \frac{{\bar{m}}_{t}}{\sqrt{{\hat{v}}_{t}}+\varepsilon },$$
where
(21)$${\bar{m}}_{t}=\left(1-\beta_{1,t}\right){\hat{g}}_{t}+\beta_{1,t+1}{\hat{m}}_{t},\;{\hat{m}}_{t}=\frac{m_{t}}{1-\prod_{i=1}^{t+1}\beta_{1i}},\;{\hat{g}}_{t}=\frac{g_{t}}{1-\prod_{i=1}^{t+1}\beta_{1i}}.$$

## 4. Results

This section investigates the experimental validation of the SMOEGE-HDL model under different aspects. Table 1 provides the overall results offered by the SMOEGE-HDL model under varying file sizes in terms of encryption (ENT), encryption memory (ENM), key size (KS), key breaking time (KBT), decryption time (DT), and decryption memory (DM). The experimental values reported that the SMOEGE-HDL model has shown enhanced encryption results under varying file sizes.

Table 2 and Figure 2 investigate the comparative results offered by the SMOEGE-HDL model with other existing methods. Based on ET and 10 kb file, the SMOEGE-HDL model has attained a lower ET of 448 s, whereas the ECC, HE, and OHE techniques have achieved increased ET of 589.958 s, 574.692 s, and 561.121 s, respectively. Concurrently, based on DT and 10 kb file, the SMOEGE-HDL method has obtained a lower DT of 72 s, whereas the ECC, HE, and OHE techniques have achieved increased DT of 113.433 s, 103.408 s, and 97.940 s, correspondingly. In parallel, based on KBT and 10 kb file, the SMOEGE-HDL model has acquired a lower KBT of 116 ms, whereas the ECC, HE, and OHE approaches have attained increased KBT of 90 ms, 94 ms, and 96 ms, correspondingly. Eventually, based on KS and 10 kb files, the SMOEGE-HDL model has attained lower KS of 44 kb, whereas the ECC, HE, and OHE methods have acquired increased KS of 20 kb, 21 kb, and 22 kb, correspondingly.

Table 3 and Figure 3 illustrate the EM and DM inspection of the SMOEGE-HDL model with other methods. The results implied that the SMOEGE-HDL model has shown enhanced results with higher EM and DM. For instance, based on EM, the SMOEGE-HDL model has shown enhanced EM of 1188 kb, whereas the ECC, HE, and OHE models have demonstrated reduced EM of 1022.26 kb, 1066.54 kb, and 1088.68 kb, respectively. In addition, based on DM, the SMOEGE-HDL technique has exhibited enhanced DM of 679 kb, whereas the ECC, HE, and OHE algorithms have established reduced DM of 510.294 kb, 526.898 kb, and 560.106 kb, correspondingly.

On the other hand, the data classification results of the SMOEGE-HDL model are tested using the Cleveland heart disease dataset [20]. It holds 303 samples with 13 features. Table 4 provides comprehensive classification outcomes of the SMOEGE-HDL model [21,22]. 

Figure 4 represents a comparative prec_n, reca_l, and spec_y results of the SMOEGE-HDL model. The results demonstrated that the SMOEGE-HDL model has shown enhanced results over existing models. With respect to prec_n, the SMOEGE-HDL model has attained a higher prec_n of 98.75%, whereas the IHDDS, J48, RT, RBF network, and NB-Tree models have obtained lower prec_n of 95.78%, 74.18%, 73.08%, 81.80%, and 75.77%, respectively. Comparatively, with respect to reca_l, the SMOEGE-HDL approach has gained a higher reca_l of 98.30%, whereas the IHDDS, J48, RT, RBF network, and NB-Tree algorithms have acquired lower reca_l of 97.03%, 72.83%, 73.93%, 82.48%, and 79.38%, respectively. Additionally, with respect to spec_y, the SMOEGE-HDL method has achieved a higher spec_y of 98.50%, whereas the IHDDS, J48, RT, RBF network, and NB-Tree techniques have obtained a lower specificity of 95.23%, 79.18%, 78%, 85.01%, and 79.72%, correspondingly.

Figure 5 signifies a comparative〖accu〗_y and F_score results of the SMOEGE-HDL method. The results established that the SMOEGE-HDL approach has shown enhanced results over existing models. With respect to 〖accu〗_y, the SMOEGE-HDL model has attained a higher〖accu〗_y of 98.50%, whereas the IHDDS, J48, RT, RBF network, and NB-Tree models have gained lower〖accu〗_y of 96.81%, 77.65%, 77.20%, 83.62%, and 80.28%, correspondingly. In addition, with respect to F_score, the SMOEGE-HDL algorithm has achieved a higher F_score of 98.25%, whereas the IHDDS, J48, RT, RBF network, and NB-Tree models have attained a lower F_score of 97.06%, 73.81%, 73.92%, 82.58%, and 77.31%, correspondingly.

The training accuracy (TRA) and validation accuracy (VLA) attained via the SMOEGE-HDL methodology on the test dataset is shown in Figure 6. The experimental result denoted the SMOEGE-HDL technique has achieved maximal values of TRA and VLA. The VLA is seemingly greater than TRA.

The training loss (TRL) and validation loss (VLL) reached via the SMOEGE-HDL algorithm on the test dataset are exhibited in Figure 7. The experimental result represented the SMOEGE-HDL algorithm has displayed the least values of TRL and VLL. Particularly, the VLL is lesser than TRL. Therefore, the proposed model exhibits superior results compared to other models [22,23,24,25,26,27,28,29,30].

## 5. Discussion

Based on the provided results, it appears that the SMOEGE-HDL classifier achieves the highest performance across all metrics, with high values for specificity, precision, recall, accuracy, and F-score. The IHDDS model also shows strong performance, although slightly lower than SMOEGE-HDL. On the other hand, the J48 algorithm, Random Tree, RBF-Network, and NB-Tree algorithm exhibit relatively lower performance across the metrics.

## 6. Conclusions

In this study, a new SMOEGE-HDL technique has been developed for security and classification in the IoT environment. In the data encryption phase, the SMOEGE approach is used to encrypt the data in the IoT environment. The SMO algorithm is utilized to achieve optimal key generation in the EGE (ElGamal Encryption) technique. This step aims to ensure secure and private communication of data in the IoT system. In the data classification phase, the HDL (Hybrid Deep-Learning) model is employed to perform the classification process. The HDL model combines deep-learning techniques with traditional machine-learning methods, leveraging the advantages of both approaches. To further enhance the classification performance of the HDL model, the Nadam optimizer is utilized in this study. The choice of optimizer plays a crucial role in training deep-learning models by adjusting the model’s weights and biases to minimize the classification error. The proposed SMOEGE-HDL method is experimentally validated, and the results are evaluated from various perspectives. This study conducts a comparative analysis to compare the performance of the SMOEGE-HDL algorithm with existing techniques. This comparative study demonstrates that the SMOEGE-HDL algorithm outperforms the existing techniques in terms of classification performance. However, the SMOEGE-HDL algorithm shows promise in enhancing the security and classification performance in the IoT environment. Computational overhead is one of the major problems in the research. The SMOEGE-HDL technique may introduce additional computational overhead due to the encryption process and the utilization of deep-learning models. Additionally, effective key management is crucial for secure communication in the IoT environment. Additionally, investigating lightweight encryption algorithms suitable for resource-constrained IoT devices can be valuable. Exploring privacy-enhancing approaches, such as differential privacy or secure multiparty computation, can be beneficial for secure communication and classification.

## Figures and Tables

**Figure 1 sensors-23-05596-f001:**
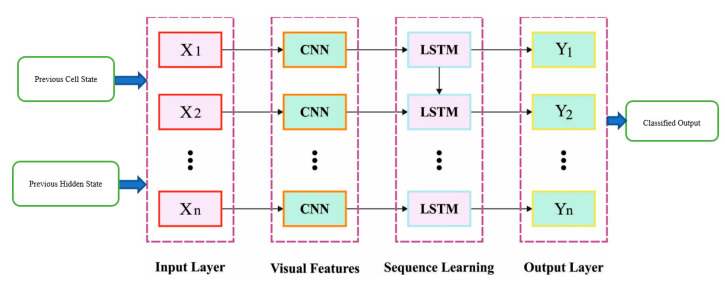
Structure of CNN-LSTM.

**Figure 2 sensors-23-05596-f002:**
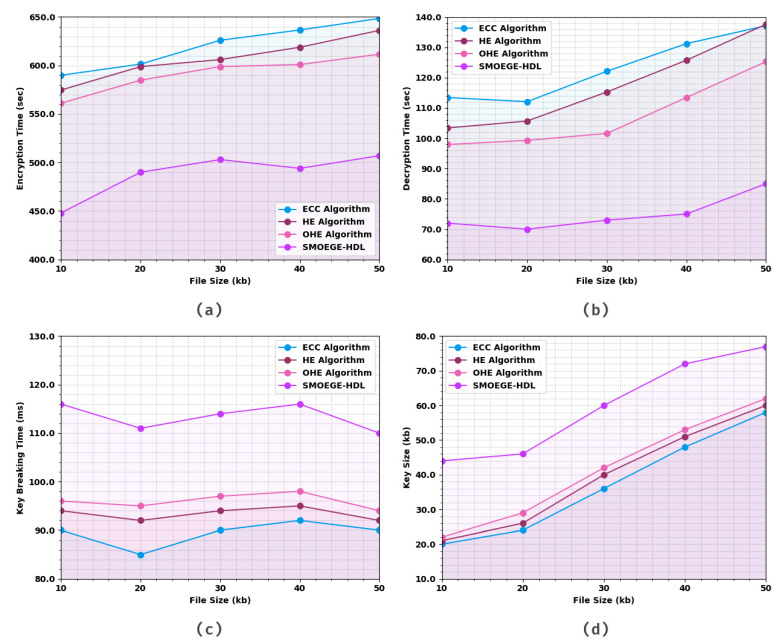
Overall result analysis of SMOEGE-HDL approach: (**a**) ET, (**b**) DT, (**c**) KBT, and (**d**) KS.

**Figure 3 sensors-23-05596-f003:**
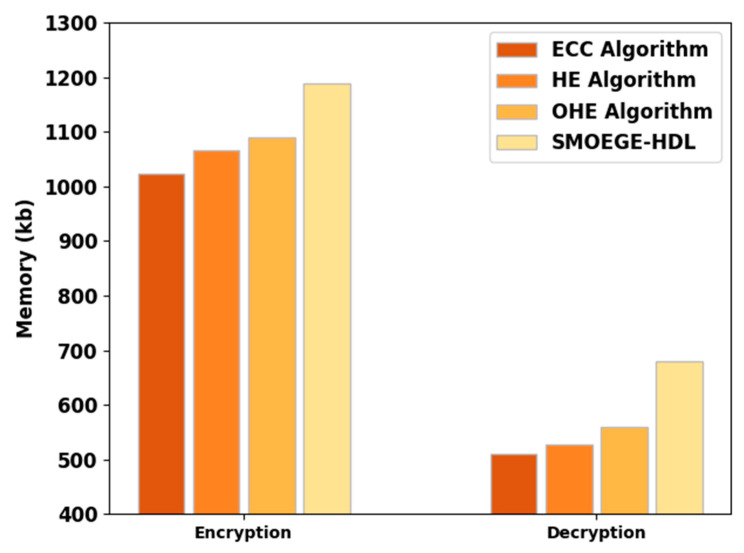
EM and DM analysis of SMOEGE-HDL technique with existing methodologies.

**Figure 4 sensors-23-05596-f004:**
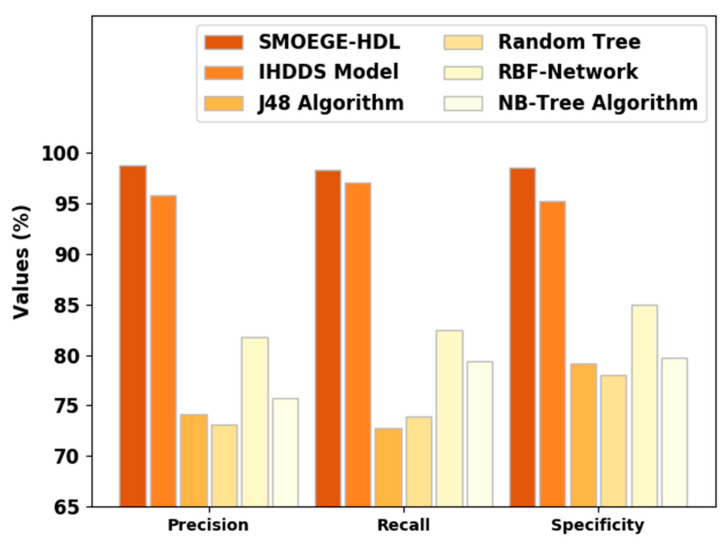
Prec_n, reca_l, and spec_y analysis of SMOEGE-HDL approach with existing methodologies.

**Figure 5 sensors-23-05596-f005:**
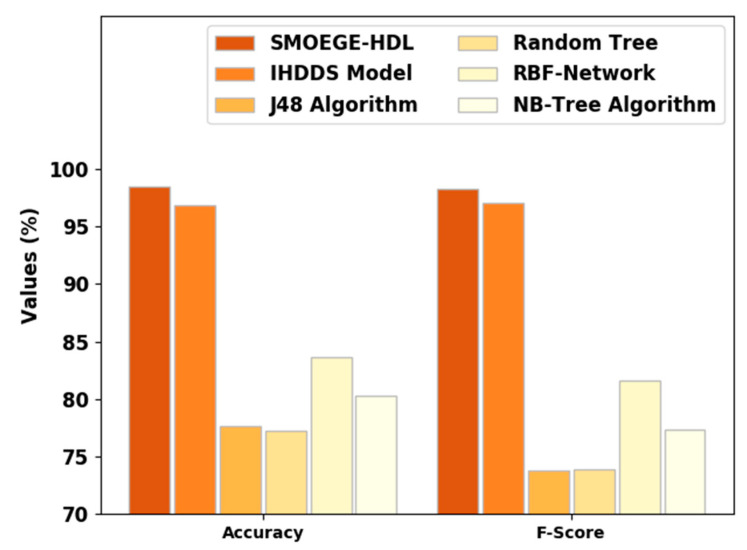
$Accu_{y}$ and $F_{score}$ analysis of SMOEGE-HDL approach with existing methodologies.

**Figure 6 sensors-23-05596-f006:**
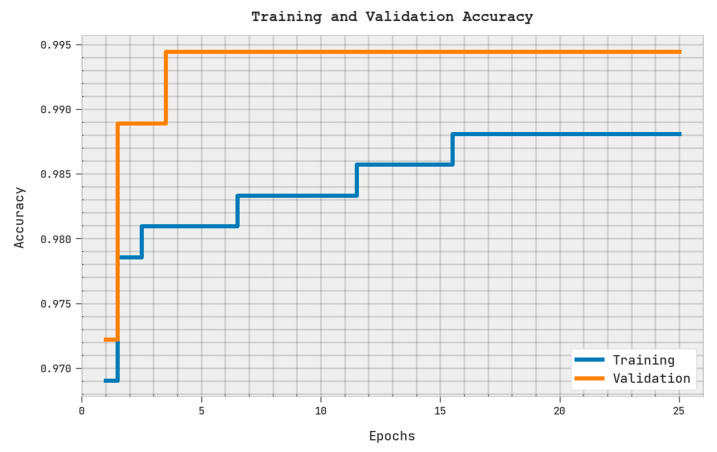
TRA and VLA analysis of SMOEGE-HDL methodology.

**Figure 7 sensors-23-05596-f007:**
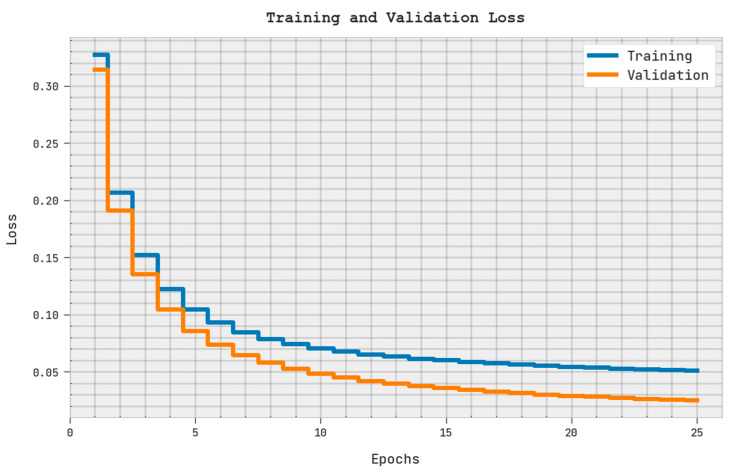
TRL and VLL analysis of SMOEGE-HDL methodology.

**Table 1 sensors-23-05596-t001:** Result analysis of SMOEGE-HDL approach.

File Size (kb)	Encryption Time (sec)	Encryption Memory (kb)	Key Size (kb)	Key Breaking Time (ms)	Decryption Time (sec)	Decryption Memory (kb)
10	448.00	1110.00	44.00	116.00	72.00	645.00
20	490.00	1168.00	46.00	111.00	70.00	647.00
30	503.00	1126.00	60.00	114.00	73.00	641.00
40	494.00	1117.00	72.00	116.00	75.00	674.00
50	507.00	1188.00	77.00	110.00	85.00	679.00

**Table 2 sensors-23-05596-t002:** Overall result analysis of SMOEGE-HDL approach with different file sizes.

Encryption Time (s)				
File Size (kb)	ECC Algorithm	HE Algorithm	OHE Algorithm	SMOEGE-HDL
10	589.958	574.692	561.121	448.00
20	601.409	598.864	584.87	490.00
30	626.005	606.073	598.864	503.00
40	636.607	618.796	600.985	494.00
50	648.481	636.183	611.586	507.00
**Decryption Time (s)**				
**File Size (kb)**	**ECC Algorithm**	**HE Algorithm**	**OHE Algorithm**	**SMOEGE-HDL**
10	113.433	103.408	97.940	72.00
20	112.066	105.687	99.307	70.00
30	122.091	115.256	101.586	73.00
40	131.205	125.737	113.433	75.00
50	137.129	137.584	125.281	85.00
**Key Breaking Time (ms)**				
**File Size (kb)**	**ECC Algorithm**	**HE Algorithm**	**OHE Algorithm**	**SMOEGE-HDL**
10	90.00	94.00	96.00	116.00
20	85.00	92.00	95.00	111.00
30	90.00	94.00	97.00	114.00
40	92.00	95.00	98.00	116.00
50	90.00	92.00	94.00	110.00
**Key Size (kb)**				
**File Size (kb)**	**ECC Algorithm**	**HE Algorithm**	**OHE Algorithm**	**SMOEGE-HDL**
10	20.00	21.00	22.00	44.00
20	24.00	26.00	29.00	46.00
30	36.00	40.00	42.00	60.00
40	48.00	51.00	53.00	72.00
50	58.00	60.00	62.00	77.00

**Table 3 sensors-23-05596-t003:** EM and DM analysis of SMOEGE-HDL technique with existing methodologies.

Memory (kb)		
**Methods**	**Encryption**	**Decryption**
ECC Algorithm	1022.26	510.294
HE Algorithm	1066.54	526.898
OHE Algorithm	1088.68	560.106
SMOEGE-HDL	1188.00	679.000

**Table 4 sensors-23-05596-t004:** Comparative analysis of SMOEGE-HDL approach with existing methodologies.

Classifiers	Specificity	Precision	Recall	Accuracy	F-Score
SMOEGE-HDL	98.50	98.75	98.30	98.50	98.25
IHDDS Model	95.23	95.78	97.03	96.81	97.06
J48 Algorithm	79.18	74.18	72.83	77.65	73.81
Random Tree	78.00	73.08	73.93	77.20	73.92
RBF-Network	85.01	81.80	82.48	83.62	81.58
NB-Tree Algorithm	79.72	75.77	79.38	80.28	77.31

## Data Availability

Not available.

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
