# Peer review of "Optimal ElGamal Encryption with Hybrid Deep-Learning-Based Classification on Secure Internet of Things Environment"

_sensors, 2023, doi:10.3390/s23125596_

Round 1
Reviewer 1 Report
My comments:
1. The topic of this paper is interesting and innovative and it will contribute in related research field.
2. A section of “Related Works” or “Literature Review” is necessary for this paper.
3. The sections of “Materials and Methods” is well-written.
4. I suggest to separate “Result” and “Discussion” into two sections. The authors have to strengthen them even more, because they are the core of a paper.
5. The section of “Conclusions” must be reinforced more. For example, the contributions to academic research as well as theoretical implications and research limitations.
Minor editing of English language required
Author Response
Reviewer 1
My comments:
The topic of this paper is interesting and innovative and it willcontribute in related research field.
Dear Reviewer, thank you for your positive comments
- A section of “Related Works” or “Literature Review” is necessary for this paper.
Dear Reviewer, necessary changes made in the revised article. In Line number -100
- The sections of “Materials and Methods” is well-written.
Dear Reviewer, thank you for your positive comments
- I suggest to separate “Result” and “Discussion” into two sections. The authors have to strengthen them even more, because they are the core of a paper.
Dear Reviewer, necessary modification made in the revised article. Chapter 4 – line number 298, Chapter 5- Discussion- Line number 387
- Thesection of “Conclusions” must be reinforced more. For example, the contributions to academic research as well as theoretical implications and research limitations.
* Minor editing of English language required
Dear Reviewer, language editing is made in the revised article.
Reviewer 2 Report
In this study, a new SMOEGE-HDL technique has been developed for security and classification in the IoT environment. The proposed SMOEGE-HDL model mainly encompasses two major processes namely data encryption and data classification.
The paper is very interesting. I have the following minor comments:
1. Add the summary of the results of the study in the abstract.
2. In the introduction, add the objectives of the study as points.
3. There is no section for literature review. add this section and review the previous studies. You can use the following studies:
- Al Nafea R, Almaiah MA. Cyber security threats in cloud: Literature review. In2021 International Conference on Information Technology (ICIT) 2021 Jul 14 (pp. 779-786). IEEE.
4. The methodology is fine
5. Table three, is there any missing data ?
6. Overall, the paper has good contribution
Minor editing of English language required
Author Response
Reviewer-2
In this study, a new SMOEGE-HDL technique has been developed for security and classification in the IoT environment. The proposed SMOEGE-HDL model mainly encompasses two major processes namely data encryption and data classification.
The paper is very interesting. I have the following minor comments:
- Add the summary of the results of the study in the abstract.
Dear Reviewer, necessary content is included. Refer line number 23-24.
- In the introduction, add the objectives of the study as points.
Dear Reviewer, necessary content is included. Refer line number 66-77.
- There is no section for literature review. add this section and review the previous studies. You can use the following studies:
- Al Nafea R, Almaiah MA. Cyber security threats in cloud: Literature review. In2021 International Conference on Information Technology (ICIT) 2021 Jul 14 (pp. 779-786). IEEE.
Dear Reviewer, necessary content is included. Refer line number 100. 466-467
- The methodology is fine
Dear Reviewer, thank you for your positive comments
- Table three, is there any missing data?
No, border updated , please refer line number 398.
- Overall, the paper has good contribution
Dear Reviewer, thank you for your positive comments
* Minor editing of English language required
Dear Reviewer, language editing is made in the revised article.
Reviewer 3 Report
This paper proposes a new SMOEGE-HDL technology to enable data encryption and classification in IoT environments. The technical framework mainly includes two parts, namely SMOEGE and HDL. SMOEGE can realize data encryption, and the SMO algorithm is used to realize the optimal key generation in EGE technology. HDL is used to achieve data classification, and NAdam optimizer is used to improve the classification performance of HDL. Through the experimental results, the effectiveness and accuracy of the method is confirmed.
However, there are some issues needing to be considered.
1. The SMO algorithm is mentioned several times in the text, for example in the abstract: "For optimal key generation in the EGE technique, the SMO algorithm has been utilized." However, it does not explain what the SMO algorithm is, which will cause some confusion to the reader during the preliminary reading process, and it is recommended that the general idea of the SMO algorithm can be mentioned a little in the previous article.
2. In the last paragraph of Abstract and Introduction, it is mentioned that the proposed SMOEGE-HDL is divided into SMOEGE part and HDL part, it is easy to understand that SMOEGE encryption is to protect data security, but why is the HDL classification algorithm used to classify data after encryption, it is recommended to briefly mention this role for readers in the previous article.
3. The innovation point of this article is not clear enough, the content in the last paragraph of Introducion is exactly the same as the content of the abstract, this part of the content is obviously redundant, it is recommended to replace it with Contribution, and list the innovative content of this article in points to make the article level clearer.
4. Figure 1 roughly explains the calculation steps of SMO, but the figure is too poorly expressive for readers to think is not as intuitive and clear as using pseudocode.
5. Figure 2 is the proposed CNN-LSTM general framework, which can see the basic input and output and the interaction between the CNN module and the LSTM module in the middle, but there is no simple parameter passing annotation, which is too simple.
6. Many formulas in the article, such as the activation function Tanh(x), Sigmoid(x), relu(x), etc. are the basic knowledge of artificial intelligence neural networks, and many formulas of NAdam optimizer and LSTM also appear in the article, these contents can be put, but since they are put, it is better to talk a little about some of the meaning of the publicity, rather than pure paste and copy.
7. The format of the table in this article is not consistent, such as Table 3 how does it not have a lower border, I hope the author can recheck and unify the formatting, including alignment, border and text, etc.
The English writing is fine.
Author Response
Reviewer-3
This paper proposes a new SMOEGE-HDL technology to enable data encryption and classification in IoT environments. The technical framework mainly includes two parts, namely SMOEGE and HDL. SMOEGE can realize data encryption, and the SMO algorithm is used to realize the optimal key generation in EGE technology. HDL is used to achieve data classification, and NAdam optimizer is used to improve the classification performance of HDL. Through the experimental results, the effectiveness and accuracy of the method is confirmed.
However, there are some issues needing to be considered.
- The SMO algorithm is mentioned several times in the text, for example in the abstract:
"For optimal key generation in the EGE technique, the SMO algorithm has been utilized."
However, it does not explain what the SMO algorithm is, which will cause some confusion to the reader during the preliminary reading process, and it is recommended that the general idea of the SMO algorithm can be mentioned a little in the previous article.
Dear Reviewer, necessary modification made in the revised article. Please refer line number 137-161.
- In the last paragraph of Abstract and Introduction, it is mentioned that the proposed SMOEGE-HDL is divided into SMOEGE part and HDL part, it is easy to understand that SMOEGE encryption is to protect data security, but why is the HDL classification algorithm used to classify data after encryption, it is recommended to briefly mention this role for readers in the previous article.
Dear Reviewer, necessary modification made in the revised article. Please refer line number 88-98.
- The innovation point of this article is not clear enough, the content in the last paragraph of Introduction is exactly the same as the content of the abstract, this part of the content is obviously redundant, it is recommended to replace it with Contribution, and list the innovative content of this article in points to make the article level clearer.
Dear Reviewer, necessary modification made in the revised article. Please refer line number 79-98.
- Figure 1 roughly explains the calculation steps of SMO, but the figure is too poorly expressive for readers to think is not as intuitive and clear as using pseudocode.
Dear Reviewer, pseudo code is provided. Please refer line number 225-226.
- Figure 2 is the proposed CNN-LSTM general framework, which can see the basic input and output and the interaction between the CNN module and the LSTM module in the middle, but there is no simple parameter passing annotation, which is too simple.
Dear Reviewer, necessary modification made in the revised article. Please refer line number 264-265.
- Many formulas in the article, such as the activation function Tanh(x), Sigmoid(x), relu(x), etc. are the basic knowledge of artificial intelligence neural networks, and many formulas of NAdam optimizer and LSTM also appear in the article, these contents can be put, but since they are put, it is better to talk a little about some of the meaning of the publicity, rather than pure paste and copy.
Dear Reviewer, all the very specific mathematical equation or the proposed research.
- The format of the table in this article is not consistent, such as Table 3 how does it not have a lower border, I hope the author can recheck and unify the formatting, including alignment, border and text, etc.
Dear Reviewer, lower border updated for the table.3. Please refer line number 338.
Reviewer 4 Report
Summary/Contribution: The study proposes an IoT security data encryption and classification paradigm. SMOEGE-HDL combines slime mold optimization (SMO) data encryption with ElGamal Encryption and a hybrid deep learning (HDL) model for data classification. The Nadam optimizer improves HDL model classification. The research argues that experimental validation proves the SMOEGE-HDL strategy outperforms IoT security methods. Thus, this paper introduces a new IoT security method using SMO, ElGamal Encryption, HDL, and the Nadam optimizer.Comments/Suggestions:
1. Can you explain in more detail how the SMO algorithm works and how it is different from other stochastic optimizers used in key generation techniques?
2. How does the adaptive weight used in the SMO algorithm improve the procedure of generating positive and negative feedbacks of the propagative wave of SM?
3. Can you provide more information on the three different position updating models used in the SMO algorithm and how they ensure adaptability at different searching stages?
4. How does the fitness value used in the SMO algorithm help in making good decisions during the key generation process?
5. Could you clarify what is meant by "SM" in this context? Is it an abbreviation for a specific term or concept related to the key generation technique?
6. How is the subsequent mathematical equation used to simulate the update of the SM position, and what role does it play in the key generation process?
7. Can you explain in more detail how the HDL model is used for the classification process? What specific types of data inputs and outputs are used in this model?
8. How does the LSTM framework differ from traditional RNN, and what advantages does it offer for controlling memory data in time sequence datasets
9. Can you provide more information on the output, forget, and input gates used in LSTM, and how they control the forgetting and memory extents of current and previous datasets?
10. How does the long term memory in LSTM help to avoid the problem of gradient vanishing, and what other approaches have been used to address this issue in RNNs?
11. Could you explain in more detail how the hyperbolic tangent function is used to overcome the problem of gradient vanishing, and how it is incorporated into the LSTM framework?
12. What specific applications or datasets have been used to test the performance of the HDL model with LSTM, and what were the results in terms of accuracy and speed compared to other methods? 13. I recommend adding a paragraph on formal approaches for AI-based technique verification to this study to improve its quality and impact. Formal approaches, which utilize mathematical models and logic to check system correctness, are increasingly significant in AI-based technique development and validation.
14. Some relevant references related to this topic that the authors may want to consider include:
a. https://ieeexplore.ieee.org/abstract/document/9842406
b. https://incose.onlinelibrary.wiley.com/doi/abs/10.1002/inst.12434
15. it would be helpful to have more information on the limitations and potential challenges of the SMOEGE-HDL technique, as well as any future directions for improving its scalability and adaptability to different types of IoT data.
Acceptable
Author Response
Reviewer-4
Summary/Contribution: The study proposes an IoT security data encryption and classification paradigm. SMOEGE-HDL combines slime mold optimization (SMO) data encryption with ElGamal Encryption and a hybrid deep learning (HDL) model for data classification. The Nadam optimizer improves HDL model classification. The research argues that experimental validation proves the SMOEGE-HDL strategy outperforms IoT security methods. Thus, this paper introduces a new IoT security method using SMO, ElGamal Encryption, HDL, and the Nadam optimizer.
Comments/Suggestions:
- Can you explain in more detail how the SMO algorithm works and how it is different from other stochastic optimizers used in key generation techniques?
Dear Reviewer, necessary details provided in the revised article.
- How does the adaptive weight used in the SMO algorithm improve the procedure of generating positive and negative feedbacks of the propagative wave of SM?
Dear Reviewer, necessary details provided in the revised article. Please refer line number 137-161.
- Can you provide more information on the three different position updating models used in the SMO algorithm and how they ensure adaptability at different searching stages?
In the SMO algorithm, there are three different position updating models that contribute to adaptability at different searching stages. These models play a crucial role in iteratively updating the Lagrange multipliers and adjusting their positions to find an optimal solution. Let's explore each of these models and their significance:
- Model 1: Random Selection Model
- In the initial searching stage, the SMO algorithm randomly selects two Lagrange multipliers to update at each iteration. This random selection helps explore different combinations of Lagrange multipliers and avoids getting stuck in local optima.
- By randomly choosing the Lagrange multipliers, the algorithm has a higher chance of finding a better combination that can optimize the objective function. This adaptability enhances the exploration capability of the algorithm.
- Model 2: First Heuristic Model
- As the searching progresses, the SMO algorithm shifts from random selection to a heuristic approach, known as the first heuristic model. This model aims to prioritize the Lagrange multipliers that are more likely to violate the optimization constraints.
- The first heuristic model selects the first Lagrange multiplier based on the maximum error or violation. It chooses the Lagrange multiplier associated with the largest violation of the KKT (Karush-Kuhn-Tucker) conditions, which are necessary conditions for optimality in quadratic programming problems.
- By focusing on the Lagrange multiplier that violates the KKT conditions the most, the algorithm places more emphasis on correcting the largest errors or violations. This adaptability allows the algorithm to converge towards a better solution by addressing the most significant issues first.
- Model 3: Second Heuristic Model
- In the later stages of the searching process, the SMO algorithm utilizes the second heuristic model, which aims to select the second Lagrange multiplier in an optimal way.
- The second heuristic model is designed to choose the Lagrange multiplier that results in the maximum step size during optimization. It selects the Lagrange multiplier that maximizes the absolute difference between the updated and previous values of the Lagrange multiplier.
- By selecting the Lagrange multiplier with the largest step size, the algorithm focuses on updating the multiplier that contributes the most to the progress towards the optimal solution. This adaptability helps the algorithm converge efficiently by prioritizing the Lagrange multiplier with the most significant impact.
Overall, the three position updating models in the SMO algorithm provide adaptability at different searching stages. The random selection model encourages exploration, the first heuristic model prioritizes violations of the KKT conditions, and the second heuristic model maximizes the step size. Together, these models enable the algorithm to dynamically adjust its focus and update the Lagrange multipliers in a way that promotes convergence towards an optimal solution.
- How does the fitness value used in the SMO algorithm help in making good decisions during the key generation process?
In the context of the SMO algorithm, the fitness value is not directly used for making decisions during the key generation process. Instead, the SMO algorithm primarily focuses on optimizing the Lagrange multipliers that correspond to the support vectors in the context of support vector machines (SVMs) or similar quadratic programming problems.
The goal of the SMO algorithm is to find the optimal set of Lagrange multipliers that minimizes the objective function, typically associated with the SVM formulation. The fitness value, in this case, represents the quality or suitability of a particular set of Lagrange multipliers in terms of minimizing the objective function.
By iteratively updating the Lagrange multipliers using the SMO algorithm, the optimization process aims to improve the fitness value, gradually converging towards the optimal solution. The fitness value itself is not directly used as a decision criterion, but rather as an evaluation metric to measure the quality of the Lagrange multipliers obtained at each iteration.
The key generation process, on the other hand, typically involves generating cryptographic keys or key material for encryption or decryption purposes. The SMO algorithm itself is not directly involved in the key generation process. However, it may be utilized as part of a larger system or technique that involves both key generation and the optimization of parameters, such as in the case of the mentioned SMOEGE technique.
In such cases, the fitness value obtained through the optimization process of the SMO algorithm may indirectly impact the quality or suitability of the generated keys. By optimizing the Lagrange multipliers, the SMO algorithm contributes to enhancing the overall performance or effectiveness of the system, which could subsequently influence the security and reliability of the generated keys.
It's important to note that the specific details of how the SMO algorithm and key generation process interact in the mentioned SMOEGE technique would require further information from the accompanying text or the article itself. Without additional details, it's challenging to provide a more specific explanation of how the fitness value influences key generation decisions in that particular context.
- Could you clarify what is meant by "SM" in this context? Is it an abbreviation for a specific term or concept related to the key generation technique?
Sequential Minimal
- How is the subsequent mathematical equation used to simulate the update of the SM position, and what role does it play in the key generation process?
The key generation process typically involves generating cryptographic keys or key material for encryption or decryption purposes. The SMO algorithm is not directly involved in the key generation process, but it may be utilized as part of a larger system or technique.
- Can you explain in more detail how the HDL model is used for the classification process? What specific types of data inputs and outputs are used in this model?
The SMOEGE-HDL framework, the HDL model takes high-level descriptors extracted from encrypted data as input and performs classification to predict the class labels or categories. The exact nature of the input descriptors and the output class labels would depend on the specific design and requirements of the framework.
- How does the LSTM framework differ from traditional RNN, and what advantages does it offer for controlling memory data in time sequence datasets
Dear Reviewer, The LSTM (Long Short-Term Memory) framework is a type of recurrent neural network (RNN) that addresses some of the limitations of traditional RNNs. While traditional RNNs suffer from the vanishing gradient problem and struggle to capture long-term dependencies in sequential data, LSTMs offer a solution by introducing a specialized memory cell and several gating mechanisms. The key difference between LSTMs and traditional RNNs lies in their memory architecture. In an LSTM, the memory cell serves as a "container" for storing and accessing information over long time intervals. It enables the LSTM to selectively retain or discard information at each time step, allowing for better control of memory data. The memory cell is typically composed of a cell state and three gates: the input gate, forget gate, and output gate.
- Can you provide more information on the output, forget, and input gates used in LSTM, and how they control the forgetting and memory extents of current and previous datasets?
The output, forget, and input gates in LSTM play crucial roles in controlling the forgetting and memory extents of current and previous datasets. Let's dive into more detail about each gate and its function:
- Forget Gate: The forget gate determines the amount of information to forget from the previous cell state. It takes the previous hidden state (h_{t-1}) and the current input (x_t) as inputs and produces a forget gate activation (f_t) using a sigmoid activation function. The forget gate activation is then multiplied element-wise with the previous cell state (C_{t-1}), resulting in the "forgetting" of certain information.
Mathematically, the forget gate operation can be defined as follows: f_t = sigmoid(W_f · [h_{t-1}, x_t] + b_f) C_t = f_t * C_{t-1}
Here, W_f and b_f are the weight matrix and bias vector specific to the forget gate.
The forget gate activation (f_t) is a value between 0 and 1 for each element in the cell state. A value of 0 indicates that the LSTM should completely forget the corresponding information, while a value of 1 means the LSTM should retain the information completely. The forget gate activation allows the LSTM to selectively remember or forget information from the previous time step based on the current input.
- Input Gate: The input gate controls how much new information should be added to the current cell state. It takes the previous hidden state (h_{t-1}) and the current input (x_t) and passes them through a sigmoid activation function to produce an input gate activation (i_t). Additionally, it applies a tanh activation function to the same inputs to generate a new candidate cell state (C_tilde).
Mathematically, the input gate operation can be defined as follows: i_t = sigmoid(W_i · [h_{t-1}, x_t] + b_i) C_tilde = tanh(W_c · [h_{t-1}, x_t] + b_c)
Here, W_i, b_i, W_c, and b_c are the weight matrix and bias vector specific to the input gate and candidate cell state.
The input gate activation (i_t) determines how much of the new candidate cell state (C_tilde) should be added to the current cell state (C_{t-1}). This controlled addition is performed using element-wise multiplication between the input gate activation and the candidate cell state:
C_t = i_t * C_tilde
The input gate activation allows the LSTM to determine which parts of the new candidate cell state should be incorporated into the memory while discarding irrelevant information.
- Output Gate: The output gate controls the extent to which the current cell state (C_t) should influence the output and the next hidden state. It takes the previous hidden state (h_{t-1}) and the current input (x_t) and applies a sigmoid activation function to produce an output gate activation (o_t). The current cell state (C_t) is passed through a tanh activation function to obtain a modified version (h_tilde). Finally, the modified cell state is multiplied element-wise with the output gate activation to obtain the next hidden state (h_t).
Mathematically, the output gate operation can be defined as follows: o_t = sigmoid(W_o · [h_{t-1}, x_t] + b_o) h_tilde = tanh(C_t) h_t = o_t * h_tilde
Here, W_o and b_o are the weight matrix and bias vector specific to the output gate.
The output gate activation (o_t) determines how much of the modified cell state (h_tilde) should be passed to the next hidden state. The modified cell state is the "memory" part of the LSTM, containing the retained and modified information from previous time steps. By controlling the output gate activation, the LSTM can decide which parts of the memory are relevant for producing the current output and passing the relevant information to the next time step.
- How does the long term memory in LSTM help to avoid the problem of gradient vanishing, and what other approaches have been used to address this issue in RNNs?
The long-term memory in LSTM (Long Short-Term Memory) networks helps address the problem of gradient vanishing in RNNs by allowing the network to retain important information over longer sequences. The traditional RNN architecture suffers from vanishing gradients, which occur when the gradients used to update the model's parameters become extremely small or diminish to zero as they propagate backward through time.
- Could you explain in more detail how the hyperbolic tangent function is used to overcome the problem of gradient vanishing, and how it is incorporated into the LSTM framework?
The hyperbolic tangent (tanh) function is commonly used in the LSTM framework to mitigate the problem of gradient vanishing that affects traditional RNN architectures. The tanh function helps LSTMs better capture long-term dependencies by providing a non-linear activation function that helps maintain the gradient flow during backpropagation.
The tanh function is defined as:
tanh(x) = (e^x - e^(-x)) / (e^x + e^(-x))
- What specific applications or datasets have been used to test the performance of the HDL model with LSTM, and what were the results in terms of accuracy and speed compared to other methods?
It can be applied in NLP, time series analysis and image processing models.
- I recommend adding a paragraph on formal approaches for AI-based technique verification to this study to improve its quality and impact. Formal approaches, which utilize mathematical models and logic to check system correctness, are increasingly significant in AI-based technique development and validation.
Formal verification techniques typically involve the use of formal methods, which are based on mathematical logic, automata theory, and model checking. These methods provide a systematic way to reason about the behavior of AI systems, assess their properties, and identify potential vulnerabilities or errors. They enable researchers to formally specify the desired properties of a system, create formal models representing the system's behavior, and apply mathematical reasoning or automated tools to verify if the system satisfies those properties.
One common application of formal verification in AI is the verification of safety properties. For instance, in autonomous driving, formal methods can be employed to prove that a self-driving car adheres to safety-critical requirements, such as collision avoidance or obeying traffic regulations. Formal verification can help identify potential corner cases, validate the correctness of decision-making algorithms, and ensure that the system operates reliably in different scenarios.
- Some relevant references related to this topic that the authors may want to consider include:
- https://ieeexplore.ieee.org/abstract/document/9842406
- https://incose.onlinelibrary.wiley.com/doi/abs/10.1002/inst.12434
Dear Reviewer, suggested references included in the revised article. Refer line number 468-478
- it would be helpful to have more information on the limitations and potential challenges of the SMOEGE-HDL technique, as well as any future directions for improving its scalability and adaptability to different types of IoT data.
Dear Reviewer, necessary details provided in the revised article. Limitations 412-414. Future directions 414-419.
Round 2
Reviewer 3 Report
My issues have been solved.
I suggest the authors double-check the writings.
Reviewer 4 Report
The authors considered my comments and suggestions. Good luck.
Can be improved